# Role of Exosomes and Their Potential as Biomarkers in Epstein-Barr Virus-Associated Gastric Cancer

**DOI:** 10.3390/cancers15020469

**Published:** 2023-01-12

**Authors:** Binnari Kim, Kyoung-Mee Kim

**Affiliations:** 1Department of Pathology, Ulsan University Hospital, University of Ulsan College of Medicine, Ulsan 44610, Republic of Korea; 2Department of Pathology and Translational Genomics, Samsung Medical Center, Sungkyunkwan University School of Medicine, Seoul 06351, Republic of Korea; 3Center of Companion Diagnostics, Samsung Medical Center, Seoul 06351, Republic of Korea

**Keywords:** exosome, EBV, gastric cancer, biomarker, immunotherapy

## Abstract

**Simple Summary:**

Exosomes are considered to be involved in the pathogenesis of various diseases, including cancer, viral infections, and autoimmune diseases. Research on exosomes is critical, since understanding the pathogenesis of diseases is crucial for their diagnosis and treatment. Epstein-Barr virus (EBV)-infected cells have been found to secrete exosomes for intercellular communication. Exosomal pathways play vital roles in the pathogenesis of EBV-related malignancies. This review aims to summarize the role of exosomes in EBV-associated gastric cancer and to serve as a basis for future diagnostic and therapeutic development.

**Abstract:**

Exosomes are a subtype of extracellular vesicles ranging from 30 to 150 nm and comprising many cellular components, including DNA, RNA, proteins, and metabolites, encapsulated in a lipid bilayer. Exosomes are secreted by many cell types and play important roles in intercellular communication in cancer. Viruses can hijack the exosomal pathway to regulate viral propagation, cellular immunity, and the microenvironment. Cells infected with Epstein-Barr virus (EBV), one of the most common oncogenic viruses, have also been found to actively secrete exosomes, and studies on their roles in EBV-related malignancies are ongoing. In this review, we focus on the role of exosomes in EBV-associated gastric cancer and their clinical applicability in diagnosis and treatment.

## 1. Introduction

Extracellular vesicles are lipid bilayer-delimited particles released from almost all cell types [1]. As they are known to transport biological information through proteins, nucleic acids, lipids, metabolites, and organelles, research on extracellular vesicles has been invigorated [1]. Extracellular vesicles exist in a wide variety of subtypes that are defined by size, biogenesis pathway, cargo, cellular source, and function; however, they are generally categorized into three types: exosomes, microvesicles, and apoptotic bodies [1]. Exosomes are endosomal in origin and are released from multivesicular bodies by exocytosis, with a size range of 30–150 nm [1]. Microvesicles, also known as ectosomes or microparticles, are of plasma membrane origin and are released from the cell surface with a size range of 30–1000 nm [1]. Apoptotic bodies originate from the plasma membrane and are released by dying cells undergoing apoptosis; they are large vesicles with diameters in the range of microns [1]. 

Exosomes are released from all cell types and play vital roles in cell-to-cell communication as carriers of genetic material [1]. The blood of healthy individuals contains more than 2000 trillion exosomes, whereas that of patients with cancer contains more than twice as much—indicating highly active intercellular communication through exosomes in cancer and their significance [2,3]. Indeed, many studies have revealed that intercellular interactions via exosomes significantly affect the tumor microenvironment, tumor angiogenesis, tumor growth, metastasis, and drug resistance [4]. 

Viruses are thought to hijack the exosomal pathway for regulating viral propagation, cellular immunity, and the microenvironment [5]. In the exosomal pathway, cytokines and signaling pathways are altered, leading to cell growth or migration of recipient cells for spreading infection [6]. Exosomes secreted from Epstein-Barr virus (EBV)-infected cells contain viral miRNAs and proteins that increase cytokine and chemokine production by target cells to alter the microenvironment [6,7,8,9,10,11,12,13,14] and significantly impact the pathogenesis of EBV-related malignancies [15,16,17]. Several studies have suggested that exosomes secreted from EBV-infected cells may affect tumorigenesis, metastasis, tumor microenvironment, immune escape, and viral physiology [18,19,20,21]. Polakovicova et al. reported that *Helicobacter pylori*, a major risk factor for gastric cancer, and EBV alter the expression of host miRNAs and cause immune evasion, epithelial–mesenchymal transition (EMT), and maintenance of infection [16]. Although the roles of exosomes in EBV-related malignancies are being actively researched, little is known about EBV-associated gastric cancer (EBVaGC) compared to B-cell lymphoma or nasopharyngeal cancer (NPC) [15]. In this review, we focused on the role of exosomes in EBVaGC and their clinical applicability in diagnosis and treatment. 

## 2. Balance between Lytic and Latent Cycles

EBV has two phases in its life cycle: lytic replication and latency [17]. A latent infection occurs first upon de novo EBV infection and exhibits one of three patterns depending on the infected cell type [17]. EBVaGC and Burkitt’s lymphoma belong to latency I; NPC and NK/T lymphoma belong to latency II; and lymphoblastoid cell lines belong to latency III [17]. In the latent state, the EBV genome exists in the nucleus as an episome, where closed circular plasmid DNA is incorporated with histones to produce only a limited number of latent viral genes [22,23]. The transfected latent viral genome is amplified once every host cell division and delivered to daughter cells along with the host genome [23]. This silent mode can facilitate persistence by minimizing exposure to the host immune system [23]. In the lytic state, all lytic genes of EBV are expressed simultaneously, resulting in an over 100-fold genome amplification and cell death [23]. The switch from latency to the lytic cycle is critical for viral persistence and the pathogenesis of EBV-related cancers; exosomes are also involved in the balance between these two modes [23]. Unlike B cells, EBV preferentially undergoes lytic replication in epithelial cells to increase the peripheral viral titer and B-cell reactivation, promoting further infection of epithelial cells through reactivated B cells [14]. The miR-200 family, which is known to form a double-negative feedback cycle with zinc finger E-box binding homeobox 1/2 (ZEB1/ZEB2), has been experimentally demonstrated to promote cell transition between the latent and lytic cycles [24,25]; these molecules are secreted from oral epithelial cells via exosomes to induce a lytic cascade in EBV-positive B cells, promoting viral transfer to the oral epithelium [26]. In conclusion, exosomes are involved in the switch from latency to the lytic cycle for viral persistence and play crucial roles in the pathogenesis of EBV-related cancers. 

## 3. EBV Infection in Normal Gastric Cells

EBV infects both B cells and epithelial cells [17]. Gastric epithelial cells can be infected by EBV-infected B cells or directly by EBV [17]. Integrins, ephrin receptor A2, and non-muscle myosin heavy chain IIA are important cofactors for EBV epithelial cell entry [27,28,29,30]. However, gastric mucosal cells are mainly thought to be infected by EBV through EBV-infected B cells rather than by direct infection [31]. EBV-infected lymphocytes come into contact with epithelial cells via integrin β1/β2 and translocate intracellular adhesion molecule-1 to the cell surface, promoting cell-to-cell contact, especially in latency type III cells [7]. Finally, viral particles are transmitted via the clathrin-mediated endocytosis pathway [32]. During this step, exosome-assisted communication between EBV-infected B cells and epithelial cells facilitates cell-to-cell contact-mediated EBV transmission [32]. When exosomal secretions from EBV-positive B cells and epithelial cells are blocked, viral transmission is partly inhibited [33]. This finding indicates that exosomes are involved in EBV transmission into epithelial cells. Moreover, when TGF-β was blocked using an antibody, the lytic cycle in B cells and viral transmission were suppressed in a dose-dependent manner, suggesting that TGF-β spontaneously released from epithelial cells promotes induction of the lytic cycle in B cells, leading to efficient viral transmission into epithelial cells [33]. After endocytosis, EBV-DNA is transported to the nucleus and undergoes circulation, where it is assembled into a circular mini-chromosome [34]; further, a chromatinization process after circulation allows for stable episome formation [34]. The two steps, circulation and chromatinization, are considered crucial for stabilizing latent infection [34]; latent infection is successfully established via extensive CpG methylation [17]. These findings demonstrate that exosomes are involved in EBV transmission into epithelial cells. The molecular processes are depicted in Figure 1. 

## 4. EBV Genes and Their Roles in EBVaGC

EBV infection is found in a small fraction of the non-neoplastic gastric mucosa both in vitro and in vivo [35], suggesting that clonal growth of EBV-infected gastric epithelial cells results in carcinoma rather than augmenting EBV infection of gastric epithelial cells per se [36,37]. DNA methylation machinery is considered to be involved in this process, transforming infected cells into clonal growth [20] and altering mRNA expression, which results in apoptosis inhibition, EMT, and immune evasion [21]. Furthermore, EBV induces exosome secretion by infected epithelial cells to alter the microenvironment in a manner favorable to the tumor [21].

EBVaGC is known to express Epstein-Barr virus-encoded small RNA (EBER) 1 and 2 (EBERs), EBV nuclear antigen 1 (EBNA1), and microRNA BamHI-A rightward transcripts (miR-BARTs), and to express approximately 40% of the latent membrane protein 2A (LMP2A) [35]. Further, the expression of small nucleolar RNA host gene 8 (SNHG8), a long non-coding RNA, is markedly increased in EBVaGC compared with that in EBV-negative gastric cancers or normal gastric mucosa [38,39,40]. Moreover, the knockdown of SNHG8 inhibits cell proliferation and colony formation, arrests the G0/G1 phase in vitro, and suppresses tumor growth in vivo [40]. Although exosomal SNHG8 has not been reported in EBVaGC, it has been found in exosomes obtained from the serum of patients with breast cancer and the breast milk of healthy mothers [41,42]. These genetic materials are believed to contribute to tumorigenesis, and their functions in EBVaGC are summarized in Table 1. In conclusion, exosomes are involved in EBV transmission into epithelial cells and alteration of the microenvironment. 

## 5. Roles of Exosomes in EBVaGC

EBV-infected cells actively release exosomes loaded with viral proteins and miRNAs [7,8,9,10,11,12,13] and EBERs are secreted primarily in complex with the protein La via exosomes [89,90,91]. A previous study characterized exosomes released from EBV-infected cells, including latency I and III cell lines [92]. Compared with the detection of a large number of miRNAs in the exosomes of a latency III cell line, miR-BART1 and miR-BART3 were detected at low levels, and no EBV-mRNA was detected in the exosomes of a latency I cell line [92]. Further, EBV infection alters exosome contents [93]. When exosomes from gastric cancer cell lines with and without artificial EBV infection were analyzed, the expression levels of CD63 and CD81 proteins were increased with EBV infection, indicating an increase in exosome delivery to the microenvironment [93]. The expression level of miR-155 in recipient epithelial cells was also increased in the presence of EBV [94].

These genetic materials are incorporated into cells via exosomes, suggesting that they might function through exosomal transfer [95,96]. Furthermore, accumulating evidence suggests that this exosomal transfer affects the phenotype of recipient cells and influences the microenvironment surrounding the infected cells [92,95]. EBV miR-BART15, which targets the *NLRP3* 3′-UTR, can be secreted from infected B cells via exosomes to inhibit the nucleotide-binding domain-like receptor protein 3 (NLRP3) inflammasome in non-infected cells, suggesting that EBV may trigger inflammasome activity [12]. EBV latent-infected cells also trigger antiviral immunity in dendritic cells through the selective release and delivery of RNA via exosomes [97]; furthermore, exosomes derived from EBV-positive gastric cancer cell lines suppress dendritic cell maturation [93]. EBER induces type I interferons and inflammatory cytokines via retinoic acid-induced gene I (RIG-I) activation, which are transported to exosomes and function similarly in EBV-infected latency III epithelial cells [95]. In oral squamous cell carcinoma, exosomes carrying EBER-1 can induce indoleamine 2,3-dioxygenase (IDO1) expression in monocyte-derived macrophages, with the help of interleukin (IL)-6 and tumor necrosis factor (TNF)-α-dependent mechanisms via the RIG-I signaling pathway [98]. They can further create an immunosuppressive microenvironment that influences the T-cell immune response [98]. IDO1, an effective immunosuppressive enzyme, is upregulated in EBVaGC, indicating the possibility of an immune evasion strategy [99]. Evidence of inflammasome responses to the EBV genome has been obtained, and this seems to be constitutive during latencies I, II, and III in B cells and epithelial cells [100]. Interferon gamma inducible protein 16 (IFI16) and cleaved interleukins have been detected in exosomes from latency I and III epithelial cells, which could be a strategy for EBV to escape host immunity [100]. Excessive stimulation and uncontrolled proliferation of EBV-infected B cells have been suggested to induce T-cell exhaustion, allowing escape from T-cell surveillance [101]. Furthermore, exosomal IL-1β may promote EBV persistence owing to its inability to recruit neutrophils [100]. In summary, EBV may trigger inflammasome activity through exosomal transfer in the pathogenesis of EBVaGC. 

## 6. EBV Genetic Material and Exosomes

A total of 22 miRNA precursors have been found in EBV, which were then processed into 40 mature miRNAs (miR-BARTs); the spliced and polyadenylated exons then form nuclear non-coding RNAs [20,102]. Four additional miRNAs are derived from BamHI fragment H rightward open reading frame 1 (BHRF-1), but these are only expressed during lytic infections [20]. In EBVaGC, most virus-derived polyadenylated transcripts are from BARTs and these are more abundant in epithelial tumor cells than in lymphoid cells [103]. EBV miR-BART can reduce the lytic replication of viruses by regulating expression of the viral genes *BZLF1* and *BRLF1*, suggesting that it may play a regulatory role between lytic and latent cycles [103]. In latent AGS cells, host cell transcription analysis by RNA-seq revealed that many downregulated genes were BART miRNA targets [102]; this was also observed when a full-length cDNA clone of one of the BART isoforms was artificially expressed in EBV-negative AGS cells [102]. EBV miR-BART cluster 1 and miR-BHRF 1–3 transferred via exosomes also silenced target gene expression in uninfected recipient cells, suggesting that exosomal miRNA transfer from EBV-infected cells to uninfected cells acts as a gene-silencing mechanism [9,16]. miR-BART7-3p shows the highest expression levels in EBVaGC tissues [59]. In NPC, miR-BART7-3p is hypothesized to play a role in chemoresistance, EMT, and metastasis by suppressing phosphatase and tensin homolog (PTEN) and suppressor of mothers against decapentaplegic (SMAD) 7 expression [104,105], but there have been no reports on the function of this miRNA in EBVaGC. miR-BART has various forms that either favorably or adversely affect tumors; however, most of these functions work in favor of tumors. This suggests a complex interaction between miR-BARTs and their targeting mRNA [20]. Further, as miR-BART functions only in EBVaGC, further functional studies are needed to show that miR-BARTs transfer via exosomes and induce substantial changes in recipient cells. 

As shown in Table 1, several roles of EBNA1 have been revealed in EBVaGC; however, the role of EBNA1 via exosomes is not well understood. High levels of EBNA1 expression have been observed in the exosomes of patients with multiple sclerosis [106], and the inflammatory cascade induced by the EBNA1-DNA complex from apoptotic EBV-positive B cells has been suggested to underlie the pathogenesis of multiple sclerosis [101]. Although further studies are needed in this regard, as EBNA1 is known to play a crucial role in regulating EBV genome persistence and cell division [107], its role via exosomes is expected to be related to this aspect.

The relationship between LMP2A and exosomes is also unclear. LMP2A is secreted in the form of exosomes, and cholesterol is considered to be involved in the trafficking and stability of LMP2A [108]. In a comprehensive proteomic analysis of LMP1 and LMP2A, these proteins were found to affect proteasome subunits, ubiquitin-specific conjugates, peptidase, and vesicle-trafficking proteins, indicating that they are also involved in regulating exosome trafficking [109]. Further, LMP2A in exosomes can be used for diagnosing NPC with high accuracy [110]. Unlike B-cell lymphoma and NPC, EBVaGC does not express LMP1. In one study, co-culture of LMP1-positive and -negative cells using the gastric cancer cell line AGS resulted in a gradual decrease in the number of LMP1-positive cells at each cell passage; further, LMP1-positive cells stimulated proliferation of the surrounding LMP1-negative cells with exosome-mediated epidermal growth factor receptor (EGFR) activation [111]. This could explain the downregulated LMP1 expression in patients with EBVaGC. Although little is known about the relationship between EBV genetic material and exosomes, exosomal miRNA transfer from EBV-infected cells to uninfected cells acts as a gene-silencing mechanism. 

## 7. Clinical Applicability of Exosomes

An ideal biomarker should have the following features: it should be stable in the body fluid for at least 24 h; its concentration should be determined quantitatively; and the measurement method should be easy, rapid, and affordable [112]. Exosomes, which reflect the donor cell characteristics, are present in almost all body fluids and can safely migrate long distances while being surrounded by lipid bilayers [113]; in particular, as the concentration of exosomes in cancer increases [3], exosomes in liquid biopsies are expected to facilitate non-invasive cancer diagnosis [113]. 

Some studies have evaluated the potential of exosomes as biomarkers using patient samples in EBV-related malignancy, but all of them have focused on NPC [114,115,116]. Extracellular vesicle-bound BART 13-3p appears to be the optimal selective marker to differentiate NPC from healthy cases, other types of cancer, and EBV-associated disease as controls [114]. The level of cyclophilin A, a member of the immunophilin family, was found to be significantly higher in both serum and exosomes from patients with NPC than from healthy individuals; further, when exosomal cyclophilin A and the antibody for EBV capsid antigen immunoglobulin A were combined, the diagnostic sensitivity and specificity were enhanced [115]. 

Although not exosomes, cell-free EBV DNAs were detected in the plasma of patients with EBVaGC and showed 71.4% sensitivity and 97.1% specificity [117]. When the plasma EBV ratio was repeatedly evaluated, a decrease or increase in plasma EBV DNA was observed after treatment and during tumor progression/relapse, respectively [117], suggesting that plasma EBV DNA monitoring could be important for patients with EBVaGC [118]. Further, candidate substances that could be potential exosomal targets, including LMP1, LMP2A, BARF1, EBV DNAs, EBV mRNAs, and EBV miRNAs, have been found in serum, saliva, and tumors, but most of these experiments were conducted on lymphoma or NPC [15,116]. Although more research is needed, these genetic products of EBV may be useful biomarkers for EBVaGC, as well as other EBV-related malignancies [21]. 

EBV-related oncoproteins and miRNAs have also been suggested as potential prognostic markers for EBVaGC. Upregulated BART20-5p expression in EBVaGC has been reported to be associated with poor recurrence-free survival [119]; further, the EBV-encoded circular RNA LMP2A plays crucial roles in inducing and maintaining the stemness phenotype [120] and is associated with distant metastasis and poor prognosis by promoting angiogenesis under hypoxia [74,120]. A high EBV copy number per genome (>10 copies) has also been reported to be correlated with programmed death ligand-1 (PD-L1) expression in cancer cells and poor disease-specific survival in EBVaGC [121]. The phosphatidylinositol 3-kinase (PI3K)/protein kinase B (Akt) pathway can be activated by LMP1 and LMP2A, resulting in resistance to chemotherapy [70]. Treatment with a PI3K inhibitor can augment the effect of chemotherapy in gastric cancer cell lines [70], suggesting that LMP1 and LMP2A are also involved in drug resistance. EBNA1 transfection reduces the phosphorylation of ataxia telangiectasia and Rad3 related (ATR) with phosphorylation of p38 mitogen-activated protein kinase (MAPK), resulting in increased susceptibility to olaparib; the same result was obtained when an ATR kinase or p38 MAPK inhibitor was used [122]. The potential diagnostic and prognostic biomarkers of EBVaGC are summarized in Table 2. 

Exosomes are being extensively studied as important biological mediators, as well as vehicles for drug delivery, because they are well tolerated and do not induce a significant immune reaction or toxicity, as observed even after repeated injections in mice [124,125,126,127,128,129,130]. One study reported that exosomes derived from phosphoantigen-expanded Vδ2-T cells can target and kill EBV-associated tumor cells through the Fas ligand and tumor necrosis factor-related apoptosis-inducing ligand (TRAIL) pathways and promote EBV antigen-specific CD4 and CD8 expansion in vitro; their antitumor activity was also confirmed in humanized mice [131]. Exosomes derived from γδ-T cells preserved their antitumor activities, such as tumor killing and T-cell promotion in the microenvironment of immunosuppressive NPC [132]. Although no experimental data have been obtained using exosomes as therapeutic agents in EBVaGC, these results suggest the potential of exosomes as therapeutic agents.

There are several limitations to using exosomes as biomarkers for EBVaGC. First, they may not be specific to a particular type of cancer and have a low sensitivity, particularly in the early stages of cancer [19]. Ultracentrifugation and density gradient centrifugation are the most common methods for isolating exosomes; however, these do not have a high purity [14]. Therefore, for exosomes to be used in clinical practice, more reliable isolation methods must be developed and sufficient validation is required. In conclusion, exosomes are expected to facilitate non-invasive cancer diagnosis and treatment through optimal isolation methods and applications in drug delivery and immune modulation. 

## 8. Immunotherapy in EBVaGC

Immunotherapy has recently become a topic of much interest for many types of malignancies. Efforts are being made to find an appropriate biomarker for selecting a patient group that is likely to respond to immunotherapy, and the most used biomarker is PD-L1 immunohistochemistry. One of the molecular characteristics of EBVaGC is the overexpression of PD-L1/PD-L2, which is expected to respond well to immunotherapy and has shown a good response in several studies [133,134]. One study reported that six stage IV EBVaGC cases exhibited partial responses after immunotherapy [133], while another study reported that nine stage IV EBVaGC cases showed partial responses (3/9, 33.3%) and stable disease (5/9, 55.6%) [134]. Two cases of overly pretreated patients with refractory metastatic EBVaGC, who experienced a significant and sustained response to pembrolizumab monotherapy, were published as case reports [135]. 

Although the exact mechanism of PD-L1 overexpression in EBVaGC has not yet been elucidated, it has been addressed in several papers [61,136,137]. One study showed that EBV-miR-BART11 and EBV-miR-BART17-3p upregulated the expression of PD-L1 in EBVaGC and NPC, resulting in the promotion of tumor immune escape [136]. Another study showed that PD-L1 expression was dramatically increased by activating Janus kinase 2 (JAK2)/signal transducer and activator of transcription 1 (STAT1)/interferon regulatory factor 1 (IRF-1) signaling in EBV-positive GC cells compared with that in EBV-negative GC cells after interferon (IFN)γ treatment [137]. In another study, transfection of miR-BART5-5p into EBV-negative GC cells induced protein inhibitor of activated STAT3 (PIAS3)/pSTAT3-dependent PD-L1 upregulation, while PD-L1 knockdown increased apoptosis and decreased cell proliferation, invasion ability, and migration [61]. However, the correlation between exosomes and PD-L1 expression in EBVaGC has not yet been reported. However, as exosomal PD-L1 plays a major role in tumor progression in several other malignancies and has attracted attention as a promising therapeutic target, it is expected to exert similar effects in EBVaGC [138]. 

## 9. Conclusions

In EBVaGC, similar to many other types of malignancies, exosomes may be involved in tumorigenesis, tumor microenvironment, immune escape, and the maintenance of viral physiology [18,19,20,21]. Further, exosomes are also considered to be involved in the balance between the lytic and latent cycles, which is important for viral persistence and the pathogenesis of EBV-related cancers [23]. 

Gastric epithelial cells can be infected through EBV-infected B cells or directly by EBV [17,31]. Integrins, ephrin receptor A2, and non-muscle myosin heavy chain IIA act as important cofactors for epithelial infection by EBV [27,28,29,30], while integrin β1/β2 and intracellular adhesion molecule-1 are involved in epithelial infection by EBV-infected B cells [7]. TGF-β spontaneously released from epithelial cells promotes induction of the lytic cycle in B cells and causes efficient viral transmission into epithelial cells [33].

Genetic materials from EBV-infected epithelial cells are released through exosomes and affect the tumor microenvironment. EBNA1 is thought to be involved in EBV genome persistence and cell division [107], LMP2A in regulation of exosome trafficking [109], miR-BART in target gene-silencing and balancing between the viral cycles [9,16,103], and EBER in modulation of the inflammatory response [95,98,99]. While IFI16 and cleaved interleukins are involved in escaping host immunity, IL-1β is thought to promote EBV persistence [100,101].

To date, there has been no paper showing the diagnostic usefulness of exosomes in EBVaGC; however, cell-free EBV DNA shows a high sensitivity and specificity in patients with EBVaGC [117]. Since a decrease or increase in plasma EBV DNA correlates with the clinical course, its usefulness in clinical practice is expected [117]. There are several papers showing that EBV-related oncoproteins and miRNAs are associated with prognosis in EBVaGC [70,74,119,120,122]. LMP2A and EBNA1 are considered to be associated with chemotherapy response [70,122], and are thought to be potential biomarkers for the diagnosis and prognosis of EBVaGC even within exosomes.

One of the molecular characteristics of EBVaGC is the overexpression of PD-L1/PD-L2, which has shown a good response for immunotherapy [133,134]. Although the exact mechanism of PD-L1 overexpression in EBVaGC has not been elucidated, EBV miR-BART11, EBV-BART17-3p, JAK2/STAT1/IRF-1 pathway, and PIAS3/pSTAT3 pathway induced by miR-BART5-5p are thought to be involved in PD-L1 overexpression in EBVaGC [61,136,137]. As exosomal PD-L1 has attracted attention as a promising diagnostic and therapeutic target in other malignancies, similar attention should be paid to its role in EBVaGC [138]. 

Compared with NPC or EBV-related B-cell lymphomas, there are fewer published articles on exosomes in EBVaGC. Therefore, further research is warranted to prove the functions of exosomes in EBVaGC. Although this review was based on a limited number of papers and information, we found that exosomes may be involved in tumorigenesis, tumor microenvironment, immune escape, and viral persistence in EBVaGC. By developing reliable isolation methods and validation studies, exosomes can be applied in clinical practice to diagnose and treat EBVaGC. 

## Figures and Tables

**Figure 1 cancers-15-00469-f001:**
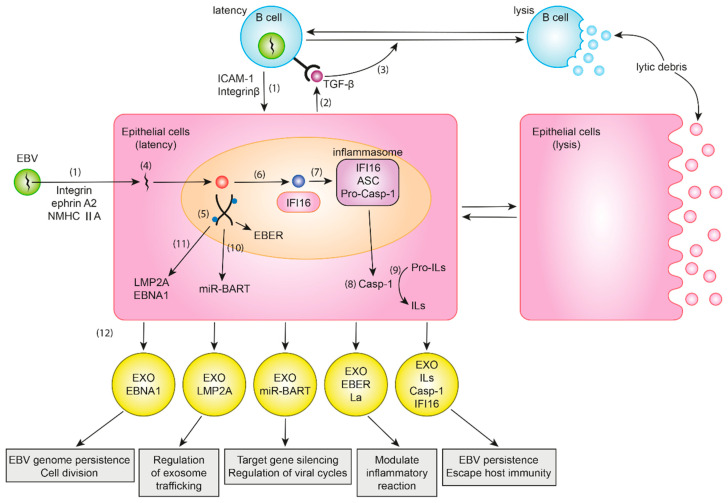
The process of epithelial EBV infection and impacts of exosomes in EBV-associated gastric cancer. (1) EBV itself or EBV-infected B cells cause EBV infection in the gastric epithelial cells. (2) TGF-β released from EBV-infected gastric epithelial cells induces expression of TGF-β receptors on B cells and (3) causes lysis of B cells leading to efficient viral transmission into epithelial cells. (4) EBV DNA transported to the nucleus undergoes (5) circulation and chromatinization. (6) Interferon gamma inducible protein 16 (IFI16) detects EBV DNA, and (7) recruits ASC and Pro-Casp-1 to form an inflammasome complex. (8) The inflammasome translocates to the cytoplasm and activates Casp-1, (9) which cleavage pro-ILs into ILs. (10) Viral transcription products (EBERs and miR-BARTs), (11) translation products (LMP2A and EBNA1), or other products (ILs, Casp-1, and IFI16) are (12) transported into recipient cells via exosomes. Abbreviations: ICAM-1, intracellular adhesion molecule-1; NMHC IIA, non-muscle myosin heavy chain IIA; IFI16, interferon gamma inducible protein 16; ASC, apoptosis-associated speck-like protein with a caspase recruitment domain; Casp-1, caspase-1; ILs, interleukins; EXO, exosome.

**Table 1 cancers-15-00469-t001:** EBV genes and their functional roles in EBV-associated gastric cancer.

Gene	Roles	Ref
*EBER*	Consistent insulin growth factor-1 expression and promotion of cell proliferation	[43]
	Increases chemoresistance and promotes cell migration	[44]
	Involved in carcinogenesis by downregulation of E-cadherin expression	[45]
	Underscores neoplastic transformation by deregulated PAR1b activation	[46]
*EBNA-1*	Promotes tumorigenicity, growth ability, immune evasion, and reduce chemosensitivity	[47]
	Induces impaired responses to DNA damage and promotes cell survival	[48]
	Maintains latent replication and persistence	[49,50]
	Leads to modification in alternative splicing profiles	[51]
	Involved in extensive methylation of the whole genome by upregulating DNMT3a	[52]
*miR-BARTs*	Involved in the initiation and development of EBVaGC	[53,54]
	miR-BART1-3p suppresses apoptosis and promotes the migration of cancer cells	[55]
	miR-BART1-3p regulates CXCL10, one of the key elements of EBVaGC	[56]
	miR-BART1-5p inhibits cell proliferation and migration	[57]
	miR-BART3-3p promotes tumorigenesis by regulating the senescence pathway	[58]
	miR-BART4-5p reduces apoptosis	[59]
	miR-BART5-3p maintains EBV latency and contributes to tumorigenesis	[60]
	miR-BART5-5p contributes to antiapoptosis, proliferation, invasion, tumor cell migration, and immune escape	[61]
	miR-BART15-3p increases apoptosis by targeting BRUCE	[62]
	miR-BART15-3p increases chemosensitivity to 5-FU	[63]
	miR-BART17-5p promotes migration and anchorage-independent growth	[64]
	miR-BART9 plays a role during carcinogenesis through EMT	[65]
	miR-BART20-5p stabilizes latent infection and regulates cell proliferation and apoptosis	[66,67]
	miR-BART16 facilitates the establishment of latent EBV infection	[68]
	miR-BART clusters I and II are involved in protection from apoptosis	[69]
*LMP2A*	Increases epithelial cell growth and differentiation with activation of the PI3K-AKT pathway	[70,71,72]
	Contributes to vasculogenic mimicry formation	[73]
	Promotes angiogenesis under hypoxia	[74]
	Contributes to tumorigenesis via phosphorylation of STAT3	[75,76]
	Inhibits apoptosis with cell cycle arrest	[77]
	Increases cell invasion and chemoresistance with the IL-6-STAT3 pathway	[44]
	Elevates cell migration and metastasis by the Notch signaling pathway	[78]
	Involved in immune evasion by activation of the Sonic Hedgehog pathway	[79,80]
	Involved in autophagy and EBV replication	[81]
	Maintains cancer stem cells in EBVaGC with NF-kB pathway	[82]
	Downregulates cell proliferation	[83,84,85]
	Downregulates expression of GCNT3 and KLF5, which promotes cell proliferation and migration	[84,86,87]
	Suppresses aryl hydrocarbon receptor pathway, which promotes cell proliferation and migration	[85]
	Promotes malignancy through epigenetic modifications by hypermethylation of AQP3 promoter	[88]
*SNHG8*	Involved in cell proliferation, colony formation, and cell growth by cell cycle regulation	[40]

**Table 2 cancers-15-00469-t002:** Potential diagnostic and prognostic biomarkers of EBVaGC.

Gene/Pathway	Specimen	Clinical Significance	Ref
EBV DNAs	Plasma	High in EBVaGC with high sensitivity and high specificity monitoring after treatment and determination of relapse	[117,118]
miR-BART20-5p	FFPE	Associated with poor recurrence-free survival in EBVaGC	[119]
Circular RNA LMP2A	FFPE, cell line	Associated with distant metastasis and poor prognosis	[74,120]
EBV-copy number/genome	FFPE, cell line	Associated with PD-L1 expression and poor disease-specific survival in EBVaGC	[121]
PI3K/Akt activation	Cell line	May induce drug resistance to chemotherapy	[70,123]
ATR/MAPK phosphorylation	Cell line	Increase susceptibility to olaparib	[122]

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
