# Peer review of "Role of Exosomes and Their Potential as Biomarkers in Epstein-Barr Virus-Associated Gastric Cancer"

_cancers, 2023, doi:10.3390/cancers15020469_

Round 1

Reviewer 1 Report

The manuscript requires extensive English language polishing by a native English speaker, who is expert in the field. I started writing my review by asking what each sentence means and avoiding over generalization. However, it seems I am unable to understand the true intent of the authors without such vigorous polishing first.

I therefore, ask that this editing be done first and a concluding statement be added to each section and then resubmitted for review. As is, I am afraid my review will not do the manuscript the deserved justice.

Examples:

Line 42-45: What does this sentence mean: “With the exosomal pathway, cytokines and signaling pathways are changed, leading to cell growth or migration of recipient cells for spread of infection”

Line 46: “change the microenvironment”: How?!

Line 46: “are also expected to have a significant impact on the pathogenesis of EBV-related malignancies”: Please avoid overgeneralization and provide specific details as to what is meant by “significant impact”.

Line 50: “Although studies about this are still being actively conducted, little is known about EBV associated gastric cancer (EBVaGC) compared to B-cell lymphoma and nasopharyngeal cancer (NPC)” : What do you mean by “this”?

Line 65: “In a follow-up study conducted by the same research team, when exosomal secretion from EBV-positive B cells was blocked, epithelial cells partly inhibited viral transmission”: This statement implies that the epithelial cells actively inhibit viral transmission, when exosomal secretion is blocked, whereas the authors of the ref (28) seem to conclude differently.

Other points:

- The EVB life cycles should be first described before referring to them.

- It would be much more informative if the paper includes a schematic diagram of what the authors wish to portray in the paper.

- A very informative review by Polakovicova et al.  (PMID: 29675003) should be described and cited

- There is no mention of the most important carcinogen of gastric cancer (Helicobacter pylori) in this review.

- The overall conclusion of the review should be more informative and include more specific details, as a take home message for the readers.

Author Response

Reviewer 1:

The manuscript requires extensive English language polishing by a native English speaker, who is expert in the field. I started writing my review by asking what each sentence means and avoiding over generalization. However, it seems I am unable to understand the true intent of the authors without such vigorous polishing first.

ANSWER: In accordance with the reviewer’s critical comments, the manuscript was sent to the professional English editing services and has been improved by an English-speaking native expert.

I therefore, ask that this editing be done first and a concluding statement be added to each section and then resubmitted for review. As is, I am afraid my review will not do the manuscript the deserved justice.

ANSWER: Thank you for your kind comments. A concluding statement was added in all sections (highlighted).

Examples:

Line 42-45: What does this sentence mean: “With the exosomal pathway, cytokines and signaling pathways are changed, leading to cell growth or migration of recipient cells for spread of infection”

ANSWER: This sentence means that virus uses the exosomal pathway, and cytokines and signaling pathways are altered, and this change can lead cell growth or migration of recipient cells for spreading of virus. We revised the sentence clearly (line 51-55).

Line 46: “change the microenvironment”: How?!

Line 46: “are also expected to have a significant impact on the pathogenesis of EBV-related malignancies”: Please avoid overgeneralization and provide specific details as to what is meant by “significant impact”.

ANSWER: Exosomes are considered to have significant impacts on tumorigenesis, metastasis, tumor microenvironment, immune escape, and maintenance of viral physiology in EBV-related malignancies. We modified the sentences to provide more specific details about “significant impact”. In addition, we added more explanations (line 51-62)

Line 50: “Although studies about this are still being actively conducted, little is known about EBV associated gastric cancer (EBVaGC) compared to B-cell lymphoma and nasopharyngeal cancer (NPC)” : What do you mean by “this”?

ANSWER: In accordance with the reviewer’s kind suggestion, the sentences were modified (line 62-64). 

Line 65: “In a follow-up study conducted by the same research team, when exosomal secretion from EBV-positive B cells was blocked, epithelial cells partly inhibited viral transmission”: This statement implies that the epithelial cells actively inhibit viral transmission, when exosomal secretion is blocked, whereas the authors of the ref (28) seem to conclude differently.

ANSWER: In accordance with the reviewer’s comments, we reviewed the reference article carefully and edited the sentence as your suggestions: “When exosomal secretion from EBV-positive B cells and epithelial cells are blocked, viral transmission was partly inhibited.” (line 100-101)

Other points:

- The EBV life cycles should be first described before referring to them.

ANSWER: We switched the orders accordingly.

- It would be much more informative if the paper includes a schematic diagram of what the authors wish to portray in the paper.

ANSWER: Thank you for your kind suggestion. We added a new and novel figure showing the process of epithelial EBV infection and impacts of exosomes in EBV-associated gastric cancer (new Figure 1 and line 365-367).

- A very informative review by Polakovicova et al.  (PMID: 29675003) should be described and cited.

ANSWER: We added the sentences from the review and added reference 16 (line 55-61). 

- There is no mention of the most important carcinogen of gastric cancer (Helicobacter pylori) in this review.

ANSWER: We added more on H. pylori with references (line 59).

- The overall conclusion of the review should be more informative and include more specific details, as a take home message for the readers.

ANSWER: In each section, we added conclusion or summary and added more information with specific details to provide a take home message for many readers (line 310-354).

Reviewer 2 Report

This systematic review supports that exosomes play a significant role in EBVaGC and can be applied in actual clinical practice for diagnosis, prognosis, and treatment in EBVaGC. The authors put forward a new point of view about relationship between exosomes and EBVaGC. The followings are comments to the authors.

1.The manuscript needs to be carefully edited by someone with expertise in technical English editing, paying particular attention to English grammar, spelling and sentence structure so that the study is clear to the readers. For example,in page 1 ,line 11 , comprise should becomprised of;

2. Please define all abbreviations in the text when used for the first time. For example, In page 1 ,line 15 EBV;

3.I suggest that the author use schematics and tables to illustrate the relationship between exosomes and EBVaGC.

Author Response

Reviewer 2:

This systematic review supports that exosomes play a significant role in EBVaGC and can be applied in actual clinical practice for diagnosis, prognosis, and treatment in EBVaGC. The authors put forward a new point of view about relationship between exosomes and EBVaGC. The followings are comments to the authors.

1.The manuscript needs to be carefully edited by someone with expertise in technical English editing, paying particular attention to English grammar, spelling and sentence structure so that the study is clear to the readers. For example, in page 1, line 11, “comprise” should be “comprised of”;

ANSWER: In accordance with the reviewer’s critical comments, the manuscript was sent to the professional English editing services and has been improved by an English-speaking native expert.

  1. Please define all abbreviations in the text when used for the first time. For example, In page 1 ,line 15 “EBV”;

ANSWER: We corrected carefully this time. Thank you or your kind comments.

  1. I suggest that the author use schematics and tables to illustrate the relationship between exosomes and EBVaGC.

ANSWER: To illustrate the relationship between exosomes and EBVaGC, we added a novel figure in the final version of the manuscript (Figure 1).

Reviewer 3 Report

This review by Binnari Kim et al. summarizes the role of exosomes and their potential as biomarkers in EBVaGC. I think the structure of the article is not very reasonable and the hierarchical logic is not clear. This review is somewhat simple, because there are not many articles on EBVaGC exosomes .

Author Response

Reviewer 3:

This review by Binnari Kim et al. summarizes the role of exosomes and their potential as biomarkers in EBVaGC. I think the structure of the article is not very reasonable and the hierarchical logic is not clear. This review is somewhat simple, because there are not many articles on EBVaGC exosomes.

ANSWER: Thank you for your critical comments. We fully agree with your opinion that there are few research papers on the role of exosomes in EBVaGC compared to B-cell lymphoma or nasopharyngeal carcinoma. Therefore, this paper may be less informative, but will be informative in summarizing the evidences that exosomes play important roles in EBVaGC and will guide future research directions. So, conclusions or summaries of each section were added in the final manuscript (highlighted) with addition of a new figure.

Round 2

Reviewer 1 Report

The English language of the manuscript is revised. However, it still requires further polishing and there remains an ambiguity in the text, as the reader can not follow a sequence of events. And therefore does not attract the reader.

Figure-1 requires a more detailed description and annotation of the sequence of events by numbering each step, in the order the authors believe takes place.

Tables 1 and 2 seem better located at the end of the manuscript, as it disrupts the text in a crude fashion.

The conclusion is still very weak and requires a more extensive summary of the presented information and a more informative conclusion, including what data is missing, thus suggesting potential future investigations in this area.

Author Response

Reviewer 1:

The English language of the manuscript is revised. However, it still requires further polishing and there remains an ambiguity in the text, as the reader can not follow a sequence of events. And therefore does not attract the reader.

ANSWER: In accordance with the reviewer’s critical comments, the manuscript was edited again by a professional English editor to enhance the readers’ interpretation. 

Figure-1 requires a more detailed description and annotation of the sequence of events by numbering each step, in the order the authors believe takes place.

ANSWER: According to the reviewer’s kind comments, we added more detailed description with numbers in the order in which the events occurred. Thank you for your critical comments.

Tables 1 and 2 seem better located at the end of the manuscript, as it disrupts the text in a crude fashion.

ANSWER: Following the reviewer’s kind comments, the tables have been moved to the end of the text.

The conclusion is still very weak and requires a more extensive summary of the presented information and a more informative conclusion, including what data is missing, thus suggesting potential future investigations in this area.

ANSWER: In accordance with the reviewer’s comments, a more extensive summary, including the main contents of the manuscript, limitations of this paper, and suggestions for future research directions has been added in the conclusion.

Reviewer 2 Report

I suggest this manuscript chould be accepted in present form.

Author Response

Thank you for your review and decision.

Reviewer 3 Report

The authors adds  corresponding content, and makes this review seem a lot richer than the previous one. 

Author Response

Thank you for your review and decision.

Round 3

Reviewer 1 Report

Figure-1 and Table-1 and 2 are missing from the revised version!! Therefore I can not tell if Figure-1 is revised